# Prescription Patterns, Recurrence, and Toxicity Rates of Adjuvant Treatment for Stage III/IV Melanoma—A Real World Single-Center Analysis

**DOI:** 10.3390/biology11030422

**Published:** 2022-03-10

**Authors:** Michèle Hoffmann, Stefanie Hayoz, Berna C. Özdemir

**Affiliations:** 1Department of Medical Oncology, Inselspital Bern, Bern University Hospital, University of Bern, 3012 Bern, Switzerland; michele.hoffmann@insel.ch; 2Swiss Group for Clinical Cancer Research, 3008 Bern, Switzerland; stefanie.hayoz@sakk.ch

**Keywords:** melanoma, adjuvant treatment, BRAF inhibitor, anti-PD1, immune checkpoint inhibitor, prescription pattern, preferences, recurrence, relapse, toxicity, real world, clinical practice

## Abstract

**Simple Summary:**

Adjuvant treatment with the immune checkpoint inhibitors (ICI) pembrolizumab or nivolumab, or the targeted therapies dabrafenib and trametinib is recommended for patients with completely resected stage III melanoma and significantly decreases recurrence risk. Currently, limited data are available on physicians’ prescription preferences regarding ICI and targeted therapies and patient outcome in clinical practice. This study investigates the real-world situation of 109 patients from the Cancer Center of the University Hospital Bern, Switzerland, with an indication for adjuvant treatment since 2018. We describe treatment patterns, recurrence, and toxicity rates under immune checkpoint inhibitors, and targeted therapies.

**Abstract:**

Approved adjuvant treatment options for stage III melanoma are the immune checkpoint inhibitors (ICI) pembrolizumab and nivolumab, and in presence of a BRAF V600E/K mutation additionally dabrafenib in combination with trametinib (BRAFi/MEKi). This study aims to describe prescription patterns and recurrence and toxicity rates of adjuvant-treated melanoma patients from the Cancer Center of the University Hospital Bern, Switzerland. One hundred and nine patients with an indication for adjuvant treatment were identified. Five (4.6%) had contraindications and, as such, were not proposed any adjuvant treatment, while 10 patients (9.2%) declined treatment. BRAF status was known for 91 (83.5%) patients. Of 40 (36.7%) patients with BRAF V600E/K melanoma, pembrolizumab was prescribed to 18 (45.0%), nivolumab to 16 (40.0%), and dabrafenib/trametinib to three (7.5%) patients. Grade 3–4 toxicity was reported in 18.9% and 16.7% of all the patients treated with pembrolizumab and nivolumab, respectively. No toxicities were observed for dabrafenib/trametinib. Thirty-eight percent of the patients treated with pembrolizumab and 40.0% of those treated with nivolumab relapsed. No relapses were reported for dabrafenib/trametinib. Prescription patterns indicate a clear preference for adjuvant ICI treatment.

## 1. Introduction

Patients with stage III melanoma are at high risk of relapse, with a melanoma specific 5-year survival ranging from 93% (IIIA) to 32% (IIID) [1]. The risk of recurrence and death is even higher in stage IV patients with resected distant metastases. About 50% of cutaneous melanomas harbor a BRAF V600E/K mutation [2].

Randomized controlled phase 3 trials have shown a significant improvement in recurrence-free survival (RFS) with adjuvant ICI or BRAFi/MEKi for stage III or IV melanoma with no evidence of disease. Importantly, these trials classified disease according to the American Joint Committee on Cancer (AJCC) 7th edition, which was replaced in 2017 by the 8th edition. 

In the Checkmate 238 trial, patients with resected stage IIIB, IIIC, and IV had a significant RFS benefit (hazard ratio, [HR] 0.65; 95% CI, 0.51 to 0.83) with nivolumab (3 mg/kg every two weeks) compared to high dose ipilimumab [3]. The EORTC 1325/Keynote 054 trial has reported a longer RFS (HR 0.57; 95% CI, 0.43 to 0.74) with pembrolizumab (200 mg every 3 weeks) compared to placebo for stage IIIA (with >1 mm lymph node metastasis), IIIB, and IIIC melanoma [4]. Similarly, the COMBI-AD trial has shown an improved RFS (HR 0.47; 95% CI, 0.42 to 0.61) for BRAF V600E/K-positive IIIA (with >1 mm lymph node metastasis), IIIB, and IIIC patients receiving dabrafenib and trametinib versus a placebo doublet [5]. 

Patients with a completely resected melanoma stage IIIA (>1 mm lymph node metastasis) or higher are eligible for adjuvant systemic treatment [6]. The indications and approval dates in Switzerland are very similar to other European countries. Nivolumab was approved for patients in stages III and IV after complete resection [3] by the European Medicines Agency (EMA) in July 2018 and by the Swiss Agency for Therapeutic Products (Swissmedic, Bern, Switzerland) in August 2018 [4]. Pembrolizumab is limited to stage IIIB-D melanoma after complete resection of lymph node metastases [4] and was approved by EMA and Swissmedic in December 2018 and February 2019, respectively. The BRAF inhibitor (BRAFi) dabrafenib in combination with the MEK inhibitor (MEKi) trametinib was approved for stage III BRAF V600E/K mutated melanoma patients after complete resection of metastases [5] by EMA and Swissmedic in August 2018. The approved treatment schedule in Switzerland for nivolumab is 240 mg every two weeks, for pembrolizumab 200 mg every three weeks, and dabrafenib (150 mg bid) and trametinib (2 mg qd) are given as continuous oral treatment. The recommended duration of treatment is one year for all adjuvant treatments.

Although head-to-head comparisons are not available and all treatment options showed a similar efficacy in clinical trials [7], the toxicity profile of BRAFi/MEKi differs significantly from that of the immune checkpoint inhibitors (ICI). The grade 3–4 toxicity rate of adjuvant dabrafenib-trametinib (41%) [5] is higher than that of nivolumab (14.4%) [3] and pembrolizumab (14.7%) [4]. While dabrafenib and trametinib toxicity is, in general, reversible and comprises pyrexia, unspecific symptoms, such as fatigue, headache and nausea, and, rarely, cardiac toxicity, immune-related adverse events (irAEs) can affect any organ, might require systemic corticosteroids or other immunosuppressive drugs, and can be long-lasting and irreversible [8]. In particular, myocarditis is a rare but potentially fatal complication of ICI [9]. 

Currently, limited data are available on physicians’ prescription preferences as well as relapse and toxicity rates of adjuvant treatments in daily clinical practice. In a recent multicenter analysis of German melanoma patients qualifying for all adjuvant treatment options, patients chose ICI more often (52.9%) than BRAFi/MEKi and the treatment selection was highly biased by physicians’ preferences [10]. According to the data from the Dutch Melanoma Treatment Registry, adjuvant ICI treatment of resected stage III/IV melanoma in the real world shows similar recurrence-free survival rates but is associated with higher toxicity rates and more frequent premature discontinuation (61%) compared to the registration trials of nivolumab (39.2%) and pembrolizumab (44.6%). 

The aim of our analysis is to describe prescription patterns, recurrence, and toxicity rates of adjuvant treatments for resected stage III/IV melanoma in daily practice, including comparing physicians’ preferences for ICI versus BRAFi/MEKi for BRAF V600E/K mutant melanoma.

## 2. Patients and Methods

### 2.1. Study Population

Medical records were used to identify melanoma patients (N = 109) with the following eligibility criteria: 18 years or older, diagnosed stages III or IV with no evidence of disease (NED) after complete tumor resection, and with an indication for adjuvant systemic treatment between 1 January 2018 and 16 June 2021. The AJCC classification 8th edition was used as the staging system. Adjuvant treatment was defined as the first systemic therapy after complete resection of melanoma. Patients receiving adjuvant treatment underwent ultrasound imaging of the regional lymph nodes alternating with 18-FDG PET-CT or CT imaging. The data cut-off date was 9 October 2021.

### 2.2. Statistical Analysis

Descriptive statistics were used to describe patient, tumor, and treatment characteristics. Treatment data were illustrated using a swimmers plot.

RFS was illustrated using Kaplan–Meier curves and RFS at 12 months was estimated with the Kaplan–Meier method at fixed time. Patients who did not have an event (recurrence or death) were censored at the date of treatment termination, and patients with ongoing treatment were censored at the date of last follow-up. The median follow-up duration was calculated with the reversed Kaplan–Meier method.

Toxicity was graded using the Common Terminology Criteria for Adverse Events (CTCAE) version 5.0. 

Statistical analyses were performed using R version 4.0.3.

## 3. Results

### 3.1. Patient and Tumour Characteristics

A total of 109 patients with an indication for adjuvant treatment were identified. The median age at the time of adjuvant treatment indication was 60 years (range 28–82 years). There was a male predominance (65.1%; N = 71). The BRAF status was known in 83.5% of patients (N = 91), a BRAF V600E/K mutation was detected in 36.7% of patients (N = 40), and an atypical BRAF mutation was found in 3.7% (N = 4) of the samples. 

Most patients (45.9%, N = 50) were classified as stage IIIC, 40.4% (N = 44) had stage IIIB, and 6.4% (N = 7) were staged as IIIA. Stage IV NED consisted of only five patients (4.6%).

The melanoma was cutaneous in 88.1% of the cases, and 11.9% had an unknown primary. The median Breslow thickness was 2.6 mm, and 39.4% of the tumors showed ulceration (Table 1).

### 3.2. Decision on Adjuvant Treatment

Of the 109 patients who had an indication for adjuvant treatment, the tumor board decided against recommending adjuvant treatment due to the patients’ comorbidities (e.g., kidney transplant recipient, autoimmune disease, dementia) in five cases (4.6%). One patient experienced rapid progression between the tumor board’s decision and patient discussion of adjuvant treatment recommendation, and, as such, was treated with first-line ipilimumab and nivolumab. Nine percent (N = 10) declined adjuvant treatment. A total of 21.1% (N = 23) of the patients wished to either start or continue their treatment at another hospital closer to their place of residence. All patients who did not receive the complete adjuvant treatment at our center for any reason were only analyzed regarding prescription type but excluded from analysis pertaining to toxicity, recurrence, and subsequent treatment (Table 1).

Eighty-three percent of patients (N = 58) began treatment within 12 weeks (84 days) of definitive surgical resection. The median duration between resected stage III/IV diagnosis and the start of adjuvant treatment was 51 days (range 14–319).

The vast majority of the patients who agreed to adjuvant treatment (N = 93) received ICI (pembrolizumab and nivolumab in 56.9% (N = 53) 39.7% (N = 37) of the patients, respectively). 

The prescription patterns for the patients treated at our center (N= 70) were very similar, with pembrolizumab in 52.9% (N = 37) and nivolumab in 42.9% (N = 30) of the cases (Figure 1A).

Of the 40 patients with BRAF V600E/K mutant melanoma, two patients declined adjuvant treatment, and one had a contraindication (severe liver cirrhosis). Three patients (7.5%) were prescribed dabrafenib/trametinib, 18 (45.0%) were prescribed pembrolizumab, and 16 (40%) were prescribed nivolumab (Figure 1B).

### 3.3. Toxicity and Recurrence Rates 

The median number of cycles received of pembrolizumab and the median duration of treatment was 15 cycles (2–19) and 11.3 months, respectively, while for nivolumab it was 20.5 cycles (2–27) and 11.6 months, respectively. All three patients receiving dabrafenib/trametinib completed one year of treatment. Grade 3–4 toxicity was documented in 18.9% and 16.7% of the patients receiving pembrolizumab or nivolumab, respectively, and led to discontinuation of pembrolizumab in 16.2% and nivolumab in 16.7% of the patients. No grade 3–4 toxicity was observed for patients under dabrafenib/trametinib (Figure 2). 

The median follow-up time was 11.3 months. The recurrence-free survival (RFS) at 12 months was 77.1% (95% CI [55.4%, 89.2%]) for nivolumab and 63.5% (95% CI [44.3%, 77.7%]) for pembrolizumab. During treatment, 32.4% (N = 12) and 20.0% (N = 6) of the patients treated with pembrolizumab and nivolumab, respectively, had disease recurrence. After stopping pembrolizumab and nivolumab, 5.4% (N = 2) and 20.0% (N = 6) of the patients, respectively, relapsed. The first site of recurrence was distant metastases in 24.3% (N = 9) and 26.7% (N = 8) of the patients treated with pembrolizumab and nivolumab, respectively. Subsequent treatment at relapse under/after pembrolizumab was most often ipilimumab/nivolumab (71.4%) and dabrafenib/trametinib (21.4%). In patients relapsing under/after adjuvant nivolumab, subsequent treatment was dabrafenib/trametinib (25.0%), ipilimumab/nivolumab (16.7%) and pembrolizumab (16.7%), respectively. None of the patients treated with dabrafenib/trametinib had relapsed at the cut-off date (Table 2 and Figure 3A,B). Two patients with distant relapse during adjuvant therapy with ICI died after subsequent therapy with dabrafenib and trametinib. One patient with distant relapse died after three lines of systemic therapy, including treatment in a clinical trial. All nine patients with local recurrence underwent tumor resection followed by systemic therapy in six cases. Of the three patients who did not receive a subsequent systemic therapy, two were BRAF wildtype and experienced grade 3–4 toxicity under adjuvant therapy with ICI. Seven out of the nine patients with local relapse had an ongoing complete remission at the cut-off date, as well as one patient who had a microwave ablation of a single hepatic metastasis.

## 4. Discussion

Our single-center analysis shows a clear preponderance of ICI as adjuvant treatment. Only 7.5% of the BRAF V600E/K mutant melanoma patients were prescribed adjuvant dabrafenib/trametinib in this setting. Given the retrospective nature of our analysis, data were not available to determine the extent to which the physicians’ preferences and patients’ choices contributed to this prescription pattern.

Lodde et al. recently reported that BRAF V600E/K patients treated in German centers preferred ICI (52.9%) over BRAFi/MEKi (47.1%) and that this choice was strongly influenced by physicians’ preferences [10]. This striking difference in prescription rate of BRAFi/MEKi between our center and the German centers suggests significant regional disparities in clinical practice possibly influenced by physicians’ preferences, local guidelines, and patient’s perceptions. It is conceivable that physicians at our center favor adjuvant ICI because ICI treatment was shown to be less efficacious in the metastatic setting if given after progression on BRAFi/MEKi [11,12]. This has been related to the changes in the tumor microenvironment and the lower CD8 T cell infiltrate in tumor tissues of patients progressing on BRAFi/MEKi [13]. More data on prescription patterns from different countries are needed to better understand the factors guiding treatment choices from physicians as well as patients. 

Several predictive biomarkers of response to ICI have been identified in advanced or metastatic melanoma. Transcriptomic analysis and immune profiling have revealed that the presence of EOMES + CD69 + CD45RO+ effector memory T cells and their gene expression profile was associated with response to ICI therapy [14]. In particular, high tumor mutation burden and interferon γ (IFΝγ) signature are enriched in responders, while no prominent mechanisms of resistance were identified in a recent multiomics analysis of patient samples [15]. The efficacy of ICI seems to be affected not only by the number and type of effector T cells but also by their spatial distribution. Higher density of CD8 T cells in close proximity (20 μm) to melanoma cells is associated with improved outcome of patients treated with ICI [16]. Moreover, the occurrence of immune-related adverse events (irAEs), especially endocrine and skin toxicity, is correlated with higher ICI efficacy as shown in several meta-analyses [17,18]. Intriguingly, emerging evidence suggest that patients who develop immune related toxicities such as vitiligo, keratitis, uveitis, and erythema nodosum under BRAFi/MEKi might have durable benefit from these therapies [19].

Besides tumor characteristics, host factors such as sex, age, dietary habits [20], and gut microbiome [21] also affect responses to ICI and BRAFi. Male and older patients seem to benefit more from ICI as compared to female or younger patients [22,23,24,25] and the likelihood of responding increases with age. This seems to be related to the higher abundance of regulatory T cells in younger patients and seems to be independent of the presence of a more complex mutational landscape [26]. Whether differences in sex hormone levels between older and younger and male and female patients, respectively, also contribute to these disparities remains unknown [27]. Experiments have shown that although melanoma growth is slower in old (>52 weeks old) compared to young (8 weeks old) mice, these tumors are more prone to metastases and resistance to BRAFi induced by aged fibroblasts [28]

To date no head-to-head comparisons of the efficacy of ICI with BRAFi/MEKi in the adjuvant setting are available. Predictive or prognostic biomarkers to help selecting between both treatment options in BRAF mutant melanoma are largely missing. 

The BRIM8 trial showed an improved disease-free survival (DFS) for stage IIC-IIIB disease but not for stage IIIC on adjuvant treatment with the BRAFi vemurafenib compared to placebo. In a retrospective analysis, a positive association was found between high CD8 T-cell infiltration and high programmed death ligand 1 (PD-L1) expression in tumor tissue and DFS. In the patients with low CD8 T cell infiltration (<1%) and PD-L1 expression (<5%), vemurafenib was associated with a significantly longer DFS as compared to placebo, suggesting a benefit particularly for patients with poor prognostic factors. However, BRAFi monotherapy is not an approved adjuvant treatment option and, as such, the clinical utility of these results remains unclear [29]. IFNγ signaling seems to provide prognostic information in stage III melanoma as shown in a preliminary report of 99 patients. The median 12-month RFS was significantly higher for IFNγ high patients independent of adjuvant treatment, yet both IFNγ low and high patients had improved RFS rates when receiving adjuvant anti-PD1 treatment [30]. In addition, assessment of lymph nodes with or without melanoma metastases has shown a lower CD4/CD8 ratio and decreasing IFNγ levels with increasing number of nodal metastases [31]. 

Despite the absence of direct comparisons, available data do not show an outcome advantage of one therapy over the other. In a retrospective multicenter analysis of 147 patients relapsing on adjuvant anti-PD1 treatment, the response rate to ipilimumab alone or in combination nivolumab was 25% and 78% to BRAFi/MEKi, respectively. If the relapse occurred off anti-PD1 treatment, these response rates increased to 40% and 90%, respectively [32]. A similar multicenter analysis of 85 patients relapsing during (22%) or after (78%) adjuvant BRAFi/MEKi has shown that 68% of the patients receive ICI as a first subsequent systemic treatment and that their outcome is similar to that of stage IV patients receiving ICI in first line with a response rate of 62% to ipilimumab/nivolumab, indicating that ICI remains an effective treatment option after adjuvant BRAFi/MEKi [33]. 

However, a prospective evaluation of the response of patients relapsing on/after adjuvant treatment to first-line treatment is currently not available. Given the clear survival advantage in the metastatic setting of first-line ipilimumab/nivolumab over BRAFi/MEKi as reported by the SECOMBIT [12] and DREAMSeq [11] trials, a randomized clinical trial comparing ICI versus BRAFi/MEKi in patients relapsing on or after adjuvant treatment is required. 

Real-world evidence shows that around half of the patients treated with ICI experience at least one grade ≥ 2 toxicity, of which 35–40% are long-lasting [8]. Endocrinopathies are among the most common immune-mediated side effects and are usually irreversible and accordingly require lifelong substitution therapy [34]. Furthermore, the few available data indicate that in addition to secondary hypogonadism in the context of hypophysitis (5.6–11%), immune-mediated primary hypogonadism due to orchitis might represent a risk of later infertility [16]. An analysis of 13 metastatic melanoma patients with testicular autopsy tissue samples showed that six of the seven men (86%) who had received ICIs had impaired spermatogenesis compared to age-matched patients who were treatment naïve (N = 6) [35]. The potential effects on female fertility are currently not known. Especially in the adjuvant setting, the potential risk of ICI-related hypogonadism, premature menopause or infertility, and their long-term consequences need to be balanced against the reduction of absolute risk of disease recurrence (rather than the relative risk reduction) and discussed with the patient as it might affect the acceptance of such prophylactic therapies [36]. 

Treatment-associated grade 3–4 toxicities occurred in 41% of the patients in treated with dabrafenib/trametinib in the COMBI-AD study, whereas none of the three patients in our cohort reported any toxicity. Given the small number of patients, our findings should be interpreted as preliminary. Among our patients, the rate of grade 3–4 toxicity and discontinuation for toxicity was higher for ICI as compared to data from the clinical trials. For nivolumab, 16.7% experienced grade 3–4 toxicity and all discontinued treatment, while in the Checkmate 238 trial the grade 3–4 toxicity rate was 14.4% and discontinuation rate for toxicity was 9.7%. In our cohort, 18.9% of the patients receiving pembrolizumab showed grade 3–4 toxicity which led to discontinuation in 16.2%, while in the EORTC1325 trial, 14.7% had grade 3–4 toxicity with a discontinuation rate of 13%. 

We report similar recurrence-free survival (RFS) rates at 12 months compared to the clinical trials. In our sample, RFS rate at 12 months was 77.1% for all stages for nivolumab compared to 72.3% in Checkmate-238 and 63.5% for pembrolizumab, compared to 75.4% in the EORTC1325/Keynote-054 trial. 

These results illustrate the potential differences in patient populations and detection and reporting of adverse events in daily practice as compared to clinical trials. Indeed, half of all melanoma patients from a Danish national cohort who had started adjuvant nivolumab discontinued treatment prematurely due to toxicity or relapse. A temporary deterioration in quality of life was also documented [17].

In our center, the tumor board recommended adjuvant treatment for the vast majority of the patients with stage III/IV NED melanoma, with only very few patients (4.6%) deemed ineligible due to severe comorbidities. The acceptance rate of adjuvant treatment by our patients was 90.8% and higher than in a German cohort reported by Lodde et al. where 76.9% opted for an adjuvant treatment [10]. More patients were prescribed pembrolizumab (52.9%) than nivolumab (42.9%). The major reason for this preference is probably the 3-week schedule of pembrolizumab, which is more convenient for the patients and the physicians. Given the very similar price and application times, cost considerations and practical aspects are not paramount. 

Most patients started adjuvant treatment in the 12 weeks after definitive surgical resection, which is in accordance with the registration trials.

Potential limitations of our study are the limited number of patients and its retrospective nature. 

Our analysis suggests that adjuvant treatment for high-risk melanoma is well accepted by physicians and patients at our center. It is, therefore, of utmost importance that patients are adequately counselled on adjuvant treatment options, as well as the expected benefit on recurrence-free survival and the potential toxicity. Patients qualifying for both ICI and BRAFi/MEKi treatment should be well informed on differences in treatment application, frequency of visits, and the risk of severe, potentially irreversible, toxicity and decisions should be made according to the patients’ preferences. 

## 5. Conclusions

Our study shows that adjuvant ICI treatment is prescribed in the vast majority of the cases, and very few of the eligible patients are prescribed BRAFi/MEKi. Patient outcomes are comparable with those from the clinical trials, with a RFS rate of 72.5% at 12 months. In the absence of a clear outcome advantage of one treatment type over the other, differences in application mode, clinic visits, and toxicity profile should be discussed with the patients.

## Figures and Tables

**Figure 1 biology-11-00422-f001:**
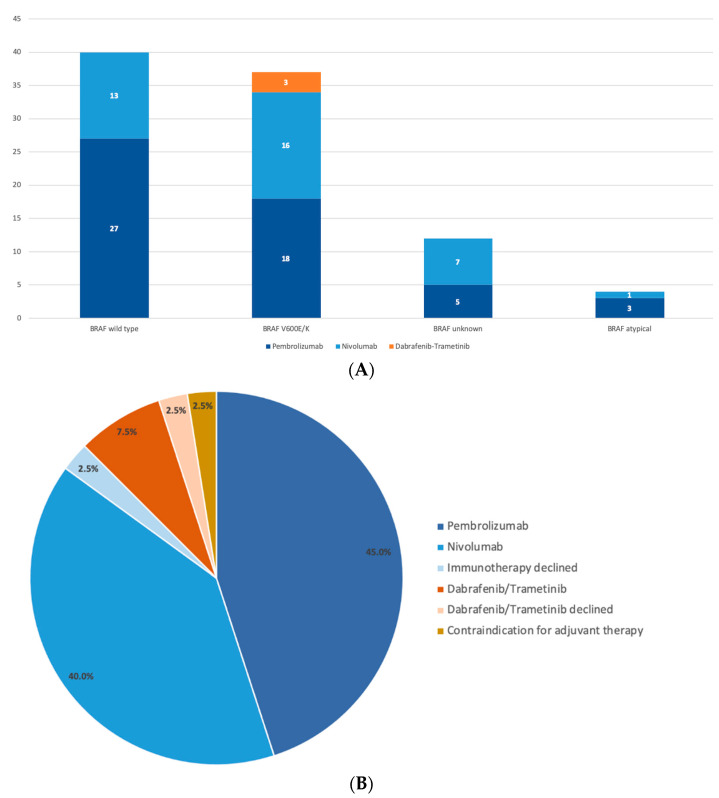
(**A**) Prescription patterns for patients who agreed on adjuvant treatment (N = 93) depicted according to their BRAF status. (**B**) Treatment decisions for BRAF V600E/K mutant melanoma (N = 40).

**Figure 2 biology-11-00422-f002:**
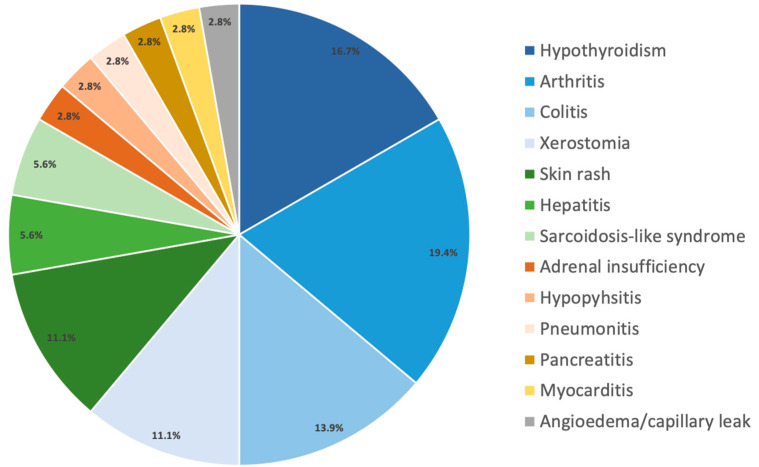
Type of all grade toxicity during or after treatment with anti-PD-1 therapy affecting 54.1% (N = 20) receiving pembrolizumab and 53.3% (N = 16) receiving nivolumab. Grade 3-4 toxicity was documented in 18.9% (N = 7) and 16.7% (N = 5) of the patients receiving pembrolizumab or nivolumab, respectively. More than one toxicity type per patient is depicted.

**Figure 3 biology-11-00422-f003:**
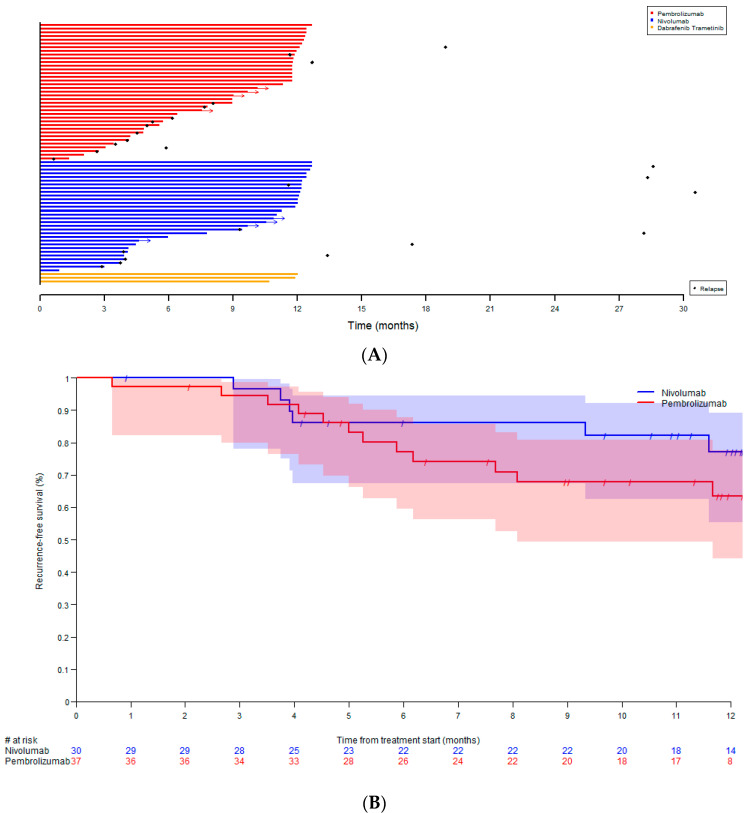
(**A**) Swimmers plot according to treatment type. (**B**) Relapse free survival according to treatment type.

**Table 1 biology-11-00422-t001:** Patient characteristics of patients with an indication for adjuvant treatment.

Patients Qualifying for Adjuvant Treatment, N	109
Sex	
Male	71 (65.1%)
Female	38 (34.9%)
Age (years)	
median (range)	60 (28–82)
BRAF status all patients/male/female	
BRAF wild type	47 (43.1%)/30/17
BRAF V600E/K	40 (36.7%)/25/15
BRAF atypical	4 (3.7%)/3/1
BRAF unknown	18 (16.5%)/13/5
Stage	
IIIA	7 (6.4%)
IIIB	44 (40.4%)
IIIC	50 (45.9%)
IIID	3 (2.8%)
IV	5 (4.6%)
Melanoma type	
Cutaneous	96 (88.1%)
Unknown origin	13 (11.9%)
Breslow thickness (mm)	
Median (range)	2.6 (0.2–12)
Tumor ulceration	
Yes	43 (39.4%)
No	66 (60.6%)
Adjuvant treatment not recommended	5 (4.6%)
Adjuvant treatment not received	11 (10.1%)
Progression before start	1 (0.9%)
Adjuvant treatment declined	10 (9.2%)
ICI	9 (8.3%)
Dabrafenib/Trametinib	1 (0.9%)
Adjuvant treatment at different hospital	23 (21.1%)
Pembrolizumab	16 (14.7%)
Nivolumab	7 (6.4%)
Dabrafenib/Trametinib	0 (0.0%)
Adjuvant treatment received at our center	70 (64.2%)
Pembrolizumab	37 (52.9%)
Nivolumab	30 (42.9%)
Dabrafenib/Trametinib	3 (4.3%)

**Table 2 biology-11-00422-t002:** Toxicity and recurrence rates according to adjuvant treatment type in our center (N = 70).

	Pembrolizumab N = 37	Nivolumab N = 30	Dabrafenib/Trametinib N = 3
Median number of cycles, range	15 (2–19)	20.5 (2–27)	12
Median duration of treatment (months), range	11.3 (1.3–12.7)	11.6 (0.8–12.7)	12.0
Toxicity all grade	20 (54.1%)	16 (53.3%)	0 (0%)
Toxicity grade 3–4	7 (18.9%)	5 (16.7%)	0 (0%)
Discontinuation			
For toxicity grade 3–4	6 (16.2%)	5 (16.7%)	0 (0%)
Recurrence	14 (37.8%)	12 (40.0%)	0 (0%)
During adjuvant treatment	12 (32.4%)	6 (20.0%)	
After stopping adjuvant treatment	2 (5.4%)	6 (20.0%)	
Site of first recurrence			
Locoregional	5 (13.5%)	4 (13.3%)	
Distant	9 (24.3%)	8 (26.7%)	
Subsequent first treatment at recurrence			N/A
Ipilimumab-Nivolumab	10 (71.4%)	2 (16.7%)	
Pembrolizumab	0 (0%)	2 (16.7%)	
Nivolumab	0 (0%)	0 (0%)	
Dabrafenib-Trametinib	3 (21.4%)	3 (25.0%)	
Encorafenib-Binimetinib	0 (0%)	1 (8.3%)	
TVEC	0 (0%)	0 (0%)	
Clinical trial	0 (0%)	1 (8.3%)	
Other (surgery, local treatment)	1 (7.1%)	3 (25.0%)	

## Data Availability

Data sets supporting reported results are archived. In case of interest, contact the corresponding author.

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
