# Peer review of "Prescription Patterns, Recurrence, and Toxicity Rates of Adjuvant Treatment for Stage III/IV Melanoma—A Real World Single-Center Analysis"

_biology, 2022, doi:10.3390/biology11030422_

Round 1
Reviewer 1 Report
The authors present a very valuable, real- world data regarding the prescription pattern, recurrence rate and toxicity rates of adjuvant therapies for melanoma patients. It was interesting that even majority of patients with BRAF mutation received ICIs. In a future study it would be nice to know what lies behind this.
I have only minor comments, as follows:
- page 2 line 59. To the best of my knowledge the indication of pembrolizumab has been extended to all stage III melanomas, who has lymphnode involvement
- Figure 2. - Number of patients having Gr 3-4 toxicities is missing from the figure legend. Please write "Sarkoidose-like Syndrome" in English.
- In the discussion part the paragraph starting with "In our center, the tumorboard..." is in a bigger font size.
Reviewer 2 Report
This is a study with a very small N, so some of its own objectives are not adequately achieved. The reasons why ICIs are preferred over BRAF/MEK inhibitors are not clearly defined or distinguished.
Therefore, no important novelty is provided.
Round 2
Reviewer 2 Report
L
This manuscript is a resubmission of an earlier submission. The following is a list of the peer review reports and author responses from that submission.
Round 1
Reviewer 1 Report
the work is potentially interesting
some changes are needed to improve the work
1. I think that in the section of materials and methods, the number of respondents should be written at the very beginning, and not to see it for the first time in the results section.
2. In the statistics section, add which program was used for the analysis
3. In addition to the sex data, it would be very nice to see the frequency of mutations in sex dependence next to the number of respondents in Table 1 because the proportion of genes by sex is not most likely the same
in the discussion the authors wrote in the role of CD4 cells and PDl however, not only CD8 cells are important for the immune response
4. It is necessary to discuss the reduced relationship of other cell populations and their capacity to produce zyrokines that play a role in controlling the immune response in melanoma, which has been previously shown and documented.
a) Decreased Interferon γ Production in CD3 + and CD3- CD56 + Lymphocyte Subsets in Metastatic Regional Lymph Nodes of Melanoma Patients. Pathol Oncol Res. 2015 Sep; 21 (4): 1109-14
5. It is necessary to comment at least, because as far as I can see it is not possible to do additional analyzes regarding NK cells in relation to immunotherapy in this paper, because survival depends on many factors including age, immune system and disease stage and needs to be supplemented with comments work
a) The role of cytokines in the regulation of NK cells in the tumor environment. Cytokine. 2019 May; 117: 30-40.
6. The response to various forms of immunotherapy has been shown and documented to depend on the state of the immune system in the past, and this should be considered as a limitation of the study.
Attenuated in vitro effects of IFN-α, IL-2 and IL-12 on functional and receptor characteristics of peripheral blood lymphocytes in metastatic melanoma patients. Cytokine. 2017, Aug; 96: 30-40.
major problems:
7. it is necessary to add data for relapse free survival depending on the age and clinical stage of the disease
Author Response
We thank the reviewer for the constructive comments and suggestions which helped to improve our manuscript. We have addressed all points raised by the reviewer.

Reviewer 2 Report
Hoffman and Colleagues presented a single center series of adjuvant treated stage III and IV ned melanoma patients. The aims of their manuscript were definined as "to inform on prescription patterns, recurrence, and toxicity rates of adjuvant treatments".
Even if there are still few real world series of melanoma adjuvant therapy in literature, the presented series account a limited number of patients in particular there were only 3 patients treated with BRAF/MEK inhibitors, these features did not allowed strenght conclusions.
They showed that in their center their attitude in prescribing adjuvant therapy was toward checkpoint inhibitors (ICI). They also showed a prevalence in pembrolizumab prescription. They also cited the multicenter analysis of German melanoma patients with the aim to explain their "clear preponderance of ICI as adjuvant treatment". But, I think that it should be scientifically correct to do a multi-center survey to define the prescribing pattern of adjuvant therapy in Swiss rather than a single-center series that reflects the opinion of the medical team of that center. Thus, I am not agree with the sentence “it is not possible to deter mine to what extent the physicians’ preferences and to what extent the patients’ choices contributed to this prescription pattern” as reported in the discussion section. It could be important to clarify these choices as well as the choice of pembrolizumab rather than nivolumab (the schedule? The duration of infusion? Pharmaeconomic issues? Other reasons?).
Regarding the recurrence pattern, even if there were 24 events, this issue could be the most interesting part of the manuscript. However, it could be more interesting to know more details.
First of all, what about the BRAF status and then the stage (III or IV ned, among the stage III); were there correlations with the extent of lymph nodes involvement? Is it possible to compare blood features (such as LDH, lymphocyte count, blood cells ratio) between patients whit and without relapse?
Finally, what about the therapeutic outcomes of subsequent treatments in relapsed patient?
Reviewer 3 Report
This retrospective analysis of adjuvant treatment of patients with stage III melanoma is nicely presented but is of limited interest due to the low patient numbers and absence of any new information.They might collaborate with other centres or highlight particular findings such as endocrinopathies
Round 2
Reviewer 2 Report
The authors tried to improve their manuscript and to replay to reviewer consideration point by point. I find the work acceptable for publication but I urge the authors to include in their manuscript the analysis of correlation between the extent of lymph nodes involvement and the comparation of blood features (such as LDH, lymphocyte count, blood cells ratio) between patients whit and without relapse.
Reviewer 3 Report
Unfortunately the paper does not contain any new information that would justify its publication. The patient numbers are too small to engage in any statements as to whether side effects differ from previous publications. The authors should consider a meta-analysis to give the study more interest